# On learning adaptive acquisition policies for undersampled multi-coil MRI reconstruction

**Tim Bakker**[*1]                                                                       T.B.BAKKER@UVA.NL
[1] *University of Amsterdam*

**Matthew Muckley**[2]                                                              MMUCKLEY@FB.COM
**Adriana Romero-Soriano**[2]                                                ADRIANARS@FB.COM
**Michal Drozdzal**[†2]                                                              MDROZDZAL@FB.COM
**Luis Pineda**[†2]                                                                       LEP@FB.COM
[2] *Facebook AI Research*

## Abstract

Most current approaches to undersampled multi-coil MRI reconstruction focus on learning the reconstruction model for a fixed, equidistant acquisition trajectory. In this paper, we study the problem of joint learning of the reconstruction model together with acquisition policies. To this end, we extend the End-to-End Variational Network with learnable acquisition policies that can adapt to different data points. We validate our model on a coil-compressed version of the large scale undersampled multi-coil fastMRI dataset using two undersampling factors: $4\times$ and $8\times$. Our experiments show on-par performance with the learnable non-adaptive and handcrafted equidistant strategies at $4\times$, and an observed improvement of more than 2% in SSIM at $8\times$ acceleration, suggesting that potentially-adaptive $k$-space acquisition trajectories can improve reconstructed image quality for larger acceleration factors. However, and perhaps surprisingly, our best performing policies learn to be explicitly non-adaptive.

**Keywords:** MRI reconstruction, undersampled multi-coil MRI, adaptive acquisition.

## 1. Introduction

Magnetic resonance imaging (MRI) is one of the best non-invasive methods for assessing soft-tissue structure in the clinic. However, widespread MRI adoption is limited due to its long acquisition times. Almost since its inception, substantial research has been done to reduce these acquisition times, yielding a variety of undersampled MRI reconstruction techniques such as parallel imaging (Sodickson and Manning, 1997; Pruessmann et al., 1999; Griswold et al., 2002), compressed sensing (Lustig et al., 2007) and deep learning (DL) (Schlemper et al., 2017; Hammernik et al., 2018; Aggarwal et al., 2018). Although the DL-based approaches have been shown to achieve the strongest results, they tend to use either fixed or random $k$-space sampling patterns that do not adapt to the data, which may be suboptimal and lead to underestimation of the maximum possible acceleration rates.

There is a substantial literature - going back decades - on designing sampling trajectories for MRI (e.g. Cao and Levin (1993); Gao and Reeves (2000); Seeger et al. (2009); Haldar

---

[*] Majority of work was done while interning at Facebook AI Research.

[†] Contributed equally.

and Kim (2019)). Since the introduction of deep learning, researchers have attempted to further improve DL-based MRI reconstruction by learning $k$-space sampling patterns from the data. These learning-based approaches result in either *non-adaptive* or *adaptive* sampling patterns – often referred to as policies. Non-adaptive policies learn a dataset specific acquisition trajectory – *e.g.*, each data point in the dataset is reconstructed following the same learnt trajectory –, while adaptive policies are conditioned on the initial $k$-space measurements and, as a result, have the potential to produce different sampling trajectories per data point. Non-adaptive policies have been shown to outperform handcrafted sampling patterns for both single-coil (Bahadir et al., 2019; Weiss et al., 2020; Huijben et al., 2020) and multi-coil (Zhang et al., 2020; Wang et al., 2021; Zibetti et al., 2021) acquisition settings. However, adaptive policies have only been devised in the single-coil setting (Zhang et al., 2019; Jin et al., 2019; Sanchez et al., 2020; Pineda et al., 2020; Bakker et al., 2020; Yin et al., 2021; Van Gorp et al., 2021), showing promising results which in some cases outperform non-adaptive ones. Learning adaptive policies for deep multi-coil MRI remains, to the best of our knowledge, largely unexplored.

In this work, we are the first to devise such a model for joint learning of 2D deep learning MRI reconstruction together with adaptive $k$-space acquisition trajectories for the more clinically relevant *multi-coil* setting. In particular, we extend recent work that learns adaptive acquisition trajectories (Yin et al., 2021) to the multi-coil scenario and enhance the End-to-End Variational Network (E2E VarNet) (Sriram et al., 2020), a standard model for multi-coil reconstruction, with the ability to learn dataset-specific as well as potentially adaptive $k$-space sampling strategies. We perform extensive evaluation on Cartesian sampling for 2D MRI using the multi-coil fastMRI knee dataset (Knoll et al., 2020b) on $4\times$ and $8\times$ acceleration. We hope our effort provides the community a point of departure for further research into adaptive multi-coil acquisition. Our experiments[1] show that:

- On the $8\times$ setup, our learned policies improve $\sim 2\%$ in SSIM over the strongest baseline, highlighting the ability of potentially-adaptive $k$-space acquisition to improve MRI reconstruction under high acceleration factors.

- On the $4\times$ setup, the gain due to $k$-space trajectory optimisation reduces, with our policies performing on-par with the strongest competing method.

- Interestingly, our top performing policies learn to be explicitly non-adaptive, suggesting that adaptivity of the $k$-space acquisition trajectories may come at the expense of over-regularising the reconstruction model.

## 2. Preliminaries

### 2.1. Background

We consider a dataset $\mathcal{D}$ of $k$-space measurements $\boldsymbol{y} \in \mathbb{C}^{N \times M}$ from which we can reconstruct MR images. In the *single-coil* setting, the reconstructed MR images can be obtained by the inverse Fourier transform $\mathcal{F}^{-1}$ as $\boldsymbol{x} = \mathcal{F}^{-1}(\boldsymbol{y})$. However, modern scanners accelerate the acquisition of the $k$-space by using multiple receiver coils that are each sensitive to different

---

1. We've made our code and pre-trained models publicly available as part of the: fastMRI repository.

regions of the anatomy, thus exploiting redundancies in $k$-space measurements (Sodickson and Manning, 1997; Pruessmann et al., 1999; Griswold et al., 2002). Hence, in the *multi-coil* setting, we define $\boldsymbol{y} \in \mathbb{C}^{N \times M \times K}$, where $K$ is the number of coils and where $\boldsymbol{y}_i \in \mathbb{C}^{N \times M}$ represents the output of a measurement by the $i$-th coil. The reconstructed MR images can then be obtained as

$$\boldsymbol{x} = \sum_{i=1}^{K} \bar{S}_i \odot \mathcal{F}^{-1}(\boldsymbol{y_i}), \tag{1}$$

where $\odot$ denotes element-wise multiplication, and $\bar{S}_i$ is the complex-conjugate of the complex-valued sensitivity map associated with each receiver coil $i$, normalised such that $\sum_{i=1}^{K} \bar{S}_i S_i = 1$. These sensitivity maps encode how sensitive each coil is to each region in the anatomy, and can be estimated in an auto-calibrating fashion by fully sampling the center of the $k$-space, also known as the auto-calibration signal (ACS) region, with each coil. The acquisition of $k$-space measurements can be further accelerated by collecting fewer measurements and reconstructing the images using a partially observed $k$-space. To simulate partial observations of the $k$-space, we introduce a Cartesian binary sampling mask $\mathbf{M}$ that selects $B < M$ measurements, including the ACS measurements, and define the *undersampled $k$-space* as $\tilde{\boldsymbol{y}}_i = \mathbf{M} \odot \boldsymbol{y}_i$, where $\odot$ denotes element-wise multiplication. Note that the same mask is applied to the measurements from all coils. The reconstructed MR images can then be obtained by leveraging the undersampled $k$-space as

$$\tilde{\boldsymbol{x}} = \sum_{i=1}^{K} \bar{S}_i \odot \mathcal{F}^{-1}(\tilde{\boldsymbol{y_i}}). \tag{2}$$

This however results in blur or aliasing effects in the reconstructed images, which can be mitigated through the use of recent deep learning models, such as the End-to-End Variational Network (E2E VarNet) (Sriram et al., 2020). In particular, the E2E VarNet takes as input the partially observed $k$-space $\tilde{\boldsymbol{y}}$ along with the sampling mask $\mathbf{M}$ decomposed into the mask of ACS measurements $\mathbf{M}_{\text{ACS}}$ and the mask of the non-ACS measurements $\mathbf{M}'$, such that $\mathbf{M} = \mathbf{M}_{\text{ACS}} + \mathbf{M}'$. The model estimates the sensitivity maps from the ACS region, and uses a cascaded neural network, $g$, to produce a high fidelity image reconstruction, $\hat{\boldsymbol{x}} = g(\tilde{\boldsymbol{y}}, \mathbf{M}_{\text{ACS}}, \mathbf{M}'; \phi)$, where $\phi$ are learnable parameters. Note that $\mathbf{M}'$ is commonly handcrafted to select equidistant measurements.

### 2.2. Problem formulation

Our goal is to adapt the sampling of measurements in $\mathbf{M}'$ to each MR slice (image). To that end, we seek a policy that, given an initial undersampled $k$-space (*e.g.*, the ACS $k$-space, $\tilde{\boldsymbol{y}}_{\text{ACS}}$), predicts which measurements to acquire next. More precisely, we aim to learn an acquisition policy $\pi(\tilde{\boldsymbol{y}}_{\text{ACS}}, \mathbf{M}_{\text{ACS}}; \theta) \to \mathbf{M}'$, parameterised by $\theta$, that selects the measurements to acquire in order to improve the image reconstruction process defined by $g$:

$$\phi^*, \theta^* = \underset{\phi, \theta}{\arg \min} \sum_{j} \mathcal{L}\left(\hat{\boldsymbol{x}}_j, \boldsymbol{x}_j\right), \tag{3}$$

where $\phi^*$ and $\theta^*$ are the optimised reconstruction and policy parameters respectively, $j$ indexes the dataset, and $\mathcal{L}$ is a loss function measuring the discrepancy between the model prediction $\hat{\boldsymbol{x}}$ and the image reconstructed from the fully sampled $k$-space $\boldsymbol{x}$.

## 3. Method

This section outlines our Policy network and details its integration with the E2E VarNet. An overview of the proposed system is depicted in Figure 1. From now on, we will use the capitalised 'Policy' to refer to our proposed model, while continuing to use 'policy' as general descriptor of 'subsampling strategy'.

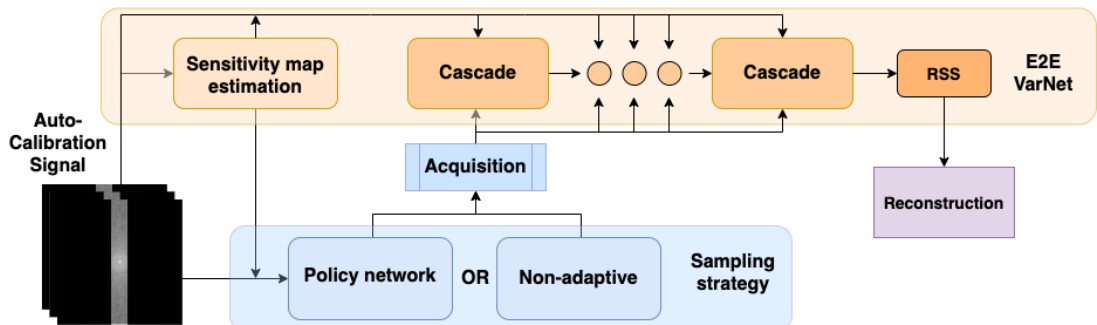

Figure 1: Overview of our system. The E2E VarNet computes sensitivity maps from the ACS, which are passed to the cascaded reconstruction model together with a subsampled $k$-space. The final reconstruction is reduced to a real-valued MR image by a root-sum-of-squares (RSS) operation. The subsampled $k$-space consists of the ACS and acquisitions suggested by a sampling strategy, which may be adaptive or non-adaptive.

### 3.1. Policy network

Our Policy is a neural network that takes as input the ACS $k$-space $\tilde{\boldsymbol{y}}_{\text{ACS}}$ and the mask of ACS measurements $\mathbf{M}_{\text{ACS}}$, and outputs a sampling probability for each measurement in $k$-space. From $\tilde{\boldsymbol{y}}_{\text{ACS}}$, we first estimate the sensitivity maps, and reconstruct a complex image $\tilde{\boldsymbol{x}}_{\text{ACS}}$ following Eq. 2. Using this image as input reduces the Policy network's size while maintaining relevant information, and allows for taking inspiration from convolutional architectures that were successfully used in the single-coil adaptive MRI literature. In particular, we employ the Policy network of Bakker et al. (2020) and extend it to handle complex-valued inputs. The network is composed of a convolutional feature extractor, followed by a fully-connected block that outputs a heatmap encoding the relative salience of each potential $k$-space measurement in the acquisition step: see Appendix C.4 for details. It remains to normalise these values and then sample $k$-space measurements. To this end, Yin et al. (2021) have shown that straight-through estimation outperforms reinforcement learning based approaches for backpropagation through discrete sampling in the single-coil setting. We thus employ their formulation, which is as follows: a non-linearity (e.g. a softplus as in Yin et al. (2021) or a sigmoid as in Bahadir et al. (2019); Zhang et al. (2020)) is first applied to ensure non-negative values. To prevent already observed $k$-space measurements in the ACS region from being sampled again, we set their corresponding probabilities to 0. The resulting vector is normalised to obtain $M$ independent realisations of a Bernoulli distribution from which to sample the measurements to be acquired. We employ rejection sampling to obtain exactly $B$ measurements on each forward pass. As a

result of the sampling, we obtain the binary mask of measurements to be acquired, $\mathbf{M}'$. The aforementioned straight-through estimation - employed during backpropagation - is realised by treating the non-differentiable sampling discretisation as a sigmoid function with slope 10. In following Yin et al. (2021), we additionally enable a fairer comparison to the multi-coil baseline of Zhang et al. (2020) (see Section 4.2), which employs such straight-through estimation as well.

### 3.2. Integrating policy network and E2E VarNet

The original E2E VarNet takes as input a partially observed $k$-space $\tilde{\boldsymbol{y}}$, a binary sampling mask of ACS measurements $\mathbf{M}_{\mathrm{ACS}}$, and a *predefined* binary sampling mask of non-ACS measurements $\mathbf{M}'$. In the adaptive acquisition setup $\mathbf{M}'$ is instead *predicted* by the policy network, so we further decompose $\tilde{\boldsymbol{y}}$ into the ACS measurements $\tilde{\boldsymbol{y}}_{\mathrm{ACS}}$ and the policy prediction $\tilde{\boldsymbol{y}}'$. As a result, our E2E VarNet takes as input $\tilde{\boldsymbol{y}}_{\mathrm{ACS}}$, $\tilde{\boldsymbol{y}}'$, $\mathbf{M}_{\mathrm{ACS}}$, and $\mathbf{M}'$. Following the original work, we first estimate sensitivity maps from the ACS measurements $\tilde{\boldsymbol{y}}_{\mathrm{ACS}}$ by means of a U-Net (Ronneberger et al., 2015). The weights of the U-Net that predict the sensitivity maps are tied between the E2E VarNet and the Policy network, such that the sensitivity maps are re-used by both networks. Then, the output of the Policy network $\mathbf{M}'$ is used to acquire $\tilde{\boldsymbol{y}}'$. As a result, we obtain a mask of observed measurements $\mathbf{M} = \mathbf{M}_{\mathrm{ACS}} + \mathbf{M}'$ and a partially observed $k$-space $\tilde{\boldsymbol{y}} = \tilde{\boldsymbol{y}}_{\mathrm{ACS}} + \tilde{\boldsymbol{y}}'$ that are used inside the E2E VarNet to produce a high fidelity MR image reconstruction. The original E2E VarNet is composed of cascaded modules that apply soft data consistency (DC) layers and refinement operations simultaneously. DC layers ensure that $k$-space predictions stay close to the observed $k$-space, while the refinement operations refine the $k$-space predictions by applying a U-Net to the corresponding complex-valued image-space representation. The cascaded and data consistency structure of the E2E VarNet offers several potential interface points with the policy model. After experimentation we chose to re-purpose the DC layers to input acquisitions made by the policy into the E2E VarNet pipeline. More precisely, we introduce hard DC layers into the E2E VarNet (Schlemper et al., 2017), which directly replace phantasised measurements in the $k$-space predictions with the observed measurements. To ensure that each cascade possesses all relevant information, we apply DC and refinement operations sequentially, rather than simultaneously. An E2E VarNet cascade now first applies a hard DC layer – which inputs acquired measurements, and restores changes to observed measurements due to a previous cascade –, followed by a refinement operation on the subsampled $k$-space $\tilde{\boldsymbol{y}}$.

## 4. Experiments

### 4.1. Data

We use the fastMRI multi-coil knee dataset (Zbontar et al., 2018) for all experiments, which contains 973 train volumes and 199 validation volumes of fully sampled $k$-space data. The test volumes are not fully sampled, and therefore cannot be used for our purposes. Our models treat every slice in a volume independently, resulting in 34,742 train slices and 7,135 validation slices to use as our dataset. For ease of experimentation, we reduce the number of coils by taking a Singular Value Decomposition and using the $K = 4$ coils with

the largest singular values (Buehrer et al., 2007). We further crop the MR slices to the center ($128 \times 128$) region of $k$-space, see Appendix C.1. To simulate clinical conditions more closely, we create the ground truth target image from the *uncompressed* $k$-space by applying a coil-wise inverse Fourier transform followed by a root-sum-of-squares (RSS) on the resulting multi-coil image representation.

### 4.2. Baselines

We compare our method to two baselines: Equispaced (or Equisdistant) and LOUPE. Equispaced masks were shown by Hammernik et al. (2018) to be a strong hand-designed subsampling strategy for deep learning-reconstructed multi-coil MRI. Such masks have been long-used in the parallel imaging literature and are a current standard for clinical 2D imaging. At the same time, they depart from random patterns favoured by compressed sensing, which is applied for single-coil reconstruction. LOUPE is a dataset-specific (*i.e.*, non-adaptive) learned strategy that aims to optimise a single subsampling mask for the entire dataset, without conditioning on initial $k$-space measurements (Bahadir et al., 2019). Recently, Zhang et al. (2020) extended LOUPE to the multi-coil setting, and we use their method as implemented by Yin et al. (2021), which employs the same normalisation and straight-through estimation used by our Policy network.

### 4.3. Training details

All models are trained to optimise SSIM (Wang et al., 2004) using Adam (Kingma and Ba, 2014) for 50 epochs with a learning rate of 0.001, decaying it by a factor 10 on epoch 40 — the default fastMRI E2E VarNet training setting. We initialise the $4\times$ acceleration experiments with the 10 lowest frequency measurements $k$-space, and acquire 22 more measurements with our models. The $8\times$ acceleration experiments are initialised with the 4 lowest frequency measurements, and we acquire 12 more, instead. This initialisation corresponds, to the ACS used in the fastMRI E2E VarNet implementation to estimate sensitivity maps[2], and is thus a natural starting point for acquisition. See Appendix C for additional details. We empirically search over several hyperparameter settings for both the reconstruction model and the policy network. For the reconstructor, we consider either 5 or 7 cascades, and either 18 or 36 channels in the first layer of the refinement modules. We also explore the heatmap non-linearity mentioned in Section 3.1 and run experiments for both the sigmoid and softplus non-linearities, using slope $\in \{1, 5, 10\}$ and $\beta \in \{0.5, 1, 5\}$, respectively. Since LOUPE employs the same normalisation and straight-through estimation as our Policy network, we also explore these non-linearities for LOUPE. Unless otherwise specified, we report the best (averaged over seeds) run under the explored hyperparameters.

### 4.4. Results

We report our main results in Table 1, where we present validation SSIM for the best performing hyperparameter setting of each model. While the Policy network performs on-par with LOUPE at $4\times$ acceleration, it outperforms the best competing method at $8\times$ accelerations by 1.89 SSIM points. To further understand the gains obtained by the Policy

---

2. As of July 30th, 2021.

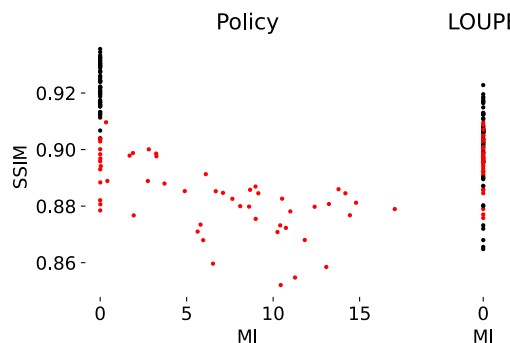

Table 1: Results on the undersampled multi-coil dataset. Policy performs on-par with the best competing model at 4× and dominates performance at 8×.

|  | Policy | LOUPE | Equispaced |
|---|---|---|---|
| 4× | **95.63** ± 0.27 | **95.61** ± 0.55 | 95.38 ± 0.03 |
| 8× | **93.26** ± 0.20 | 91.37 ± 0.67 | 91.30 ± 0.06 |

Figure 2: SSIM as a function of policy mutual information (MI), for the 8× setting. Each dot is a single model. Red: softplus; black: sigmoid.

model, we inspect the adaptivity of the learned policies by plotting the SSIM score as a function of the mutual information (MI) between probability masks and images of all learned policies (Policy and LOUPE) — see Figure 2 for the 8× acceleration results. We plot the policies that use the sigmoid non-linearity in black, and policies that use the softplus non-linearity in red. Surprisingly, we observe that the best performing Policy models exhibit zero mutual information, denoting no adaptivity — *i.e.*, the distribution of actions is constant for all data points in our dataset. We observe that early in the learning process all policies are adaptive and that some of them *learn to be non-adaptive*. Moreover, we observe that all sigmoid-based Policies end up being non-adaptive while the majority of the softplus-based Polices converge to adaptive strategies, suggesting that this non-linearity plays a crucial role in learning adaptivity. We hypothesise that it may be easier for Policies using the sigmoid non-linearity to learn non-adaptive strategies – which requires learning to ignore their input – given the saturation of the function for both very large positive and negative values. In contrast, the softplus non-linearity only saturates for large negative values, while the model is simultaneously encouraged to assign positive values to at least $B$ actions when sampling from the distribution. Figure 3 displays some examples of image reconstructions and learnt subsamplings for the 8× acceleration. The sigmoid-based Policy chooses a subset of actions to sample from with equal probability, whereas LOUPE appears to yield very-nearly sparse probabilities; exhibiting probability close to 1 for a limited set of actions, while most actions end up with a probability near 0. The more adaptive softplus-based Policy assigns less regular probability values, and seems to favour the center region less than both its sigmoid-based counterpart and LOUPE. Appendix D.2 contains additional qualitative results.

### 4.4.1. Discussion

In this subsection, we outline three hypotheses as to why the best performing policies learn to be non-adaptive. The first one is model *generalisation*: the model may learn a strong adaptive Policy on the training set that does not generalise to the validation set. However,

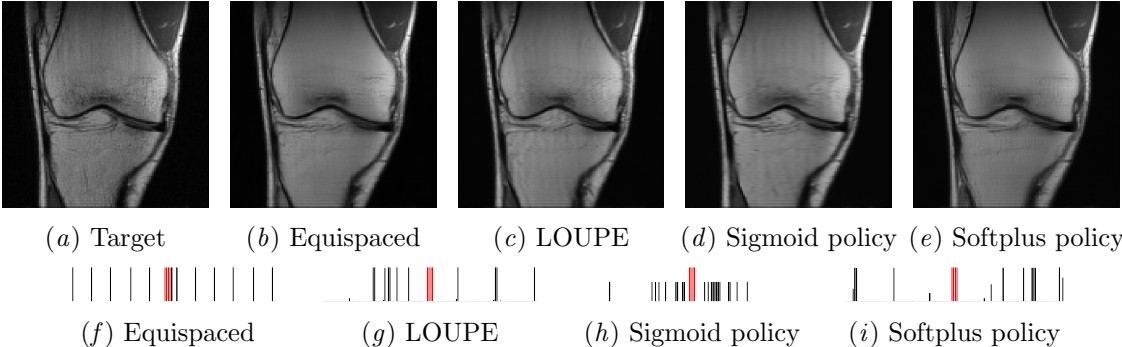

(a) Target   (b) Equispaced   (c) LOUPE   (d) Sigmoid policy   (e) Softplus policy

(f) Equispaced   (g) LOUPE   (h) Sigmoid policy   (i) Softplus policy

Figure 3: Qualitative results for 8×: (a) Ground truth, (b-e) reconstructions, and (f-g) visualisations of subsampling policies. Each policy depicts 128 probabilities; one per potential $k$-space measurement. The ACS region (red) has probability 1.

we examine the Policy adaptivity on the training set in Appendix D.1.2, and find no evidence that the lack of adaptivity is caused by overfitting the train data. The second hypothesis is connected to *amortisation* of the parameters of the reconstruction model over the large training dataset. Adaptivity can act as a regulariser on the reconstruction model, since a single reconstruction model needs to reconstruct MR images following multiple different $k$-space sampling patterns. In Appendix D.1.3, we report results for systems trained with reconstruction models of increasing capacities, and assess whether leveraging higher capacity reconstructors can help achieve Policy adaptivity. Although the higher capacity models lead to slight improvements in terms of SSIM, the best models are still not adaptive. Finally, we hypothesise that the estimation of sensitivity maps – a significant difference between single-coil and multi-coil reconstruction pipelines – may affect the adaptivity of the joint model. In the E2E VarNet, sensitivity maps are estimated independently per slice, and this may enable a form of adaptivity beyond the mask selection that we have explored here. However, the interaction between the sensitivity maps and the acquisition policies requires further investigation and is left as future work (see Appendix B). We display some learned sensitivity maps for all models in Appendix D.2.3.

## 5. Conclusion

We have explored the problem of jointly optimising an adaptive sampling strategy and a deep reconstruction model for multi-coil 2D MRI. To this end, we have proposed the first method for integrating a policy network with the E2E VarNet reconstruction model, and evaluated it on the large-scale, multi-coil fastMRI knee dataset. Our Policy networks outperform previous learning based non-adaptive approaches as well as the handcrafted equispaced masks at 8× accelerations. Interestingly however, the Policies learn to be explicitly non-adaptive. The main limitation of our work is the analysis on coil-compressed (128 × 128) $k$-space acquisitions, rather than the full data. The relevance of our work for the original, un-cropped data should be empirically verified by future research.

## Acknowledgments

T. Bakker's PhD work is partially supported by the 'Efficient Deep Learning' (EDL, https://efficientdeeplearning.nl) research programme, which is financed by the Dutch Research Council (NWO) domain Applied and Engineering Sciences (TTW).

We are grateful to the Weights&Biases team (Biewald, 2020) for providing their experiment tracking software. We would also like to thank Edward Smith for useful discussions.

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
