# OpenReview forum: "On learning adaptive acquisition policies for undersampled multi-coil MRI reconstruction"
_MIDL.io/2022/Conference — MIDL 2022_

### Official Review · Reviewer_rHh8 · 2022-01-21

**Confidence:** 5
**Preliminary Rating:** 2
**Recommendation:** Poster

**Summary:**

This paper describes a new method for MRI sampling design.  The paper addresses an interesting problem, but the literature review is poor (missing many relevant prior contributions), the explanation of the inverse problem seems to lack awareness of the past 20 years of parallel imaging research, the method appears to be incremental (without clear conceptual innovation), and the results are not impressive or good enough to be used in practical scenarios.

**Strengths:**

The paper is mostly easy to read (well written) and addresses an interesting and important problem.  The observation that non-adaptive policies are better than adaptive policies is interesting, although I'm not completely convinced that this is fundamental rather than a limitation of the narrow investigation.

**Weaknesses:**

"Over the last three decades, substantial research has been done to reduce these acquisition times" -->  More like the last 4+ decades.  Accelerated MRI image reconstruction has been around since the 1980s, and there were fast imaging methods like EPI in the late 1970s.

"In recent years, researchers have attempted to further improve DL-based MRI reconstruction by learning k-space sampling patterns from the data."  --> Optimization of sampling patterns is much older than this, but the paper seems to be unaware of a lot of the relevant references:

Cao Y, Levin DN. Feature-recognizing MRI. Magnetic resonance in medicine. 1993 Sep;30(3):305-17.

Gao Y, Reeves SJ. Optimal k-space sampling in MRSI for images with a limited region of support. IEEE transactions on medical imaging. 2000 Dec;19(12):1168-78.

Seeger M, Nickisch H, Pohmann R, Schölkopf B. Optimization of k-space trajectories for compressed sensing by Bayesian experimental design. Magnetic Resonance in Medicine: An Official Journal of the International Society for Magnetic Resonance in Medicine. 2010 Jan;63(1):116-26.

Haldar JP, Kim D. OEDIPUS: An experiment design framework for sparsity-constrained MRI. IEEE transactions on medical imaging. 2019 Feb 1;38(7):1545-58.

"The reconstructed MR images can then be obtained by leveraging the undersampled k-space as [Equation 2]" --> This is not how anyone does multi-channel image reconstruction with undersampled data, this formula only makes sense for Nyquist-sampled data and is widely understood to be naive and inadequate for undersampled data.  This description makes it seem like there is a lack of familiarity with the topic.

"Our goal is to adapt the sampling of measurements in M' to each MR slice (image)."  --> It's not clear if the authors are aware that Cartesian sampling is very different for 2D and 3D MR images.  The authors seem to be considering a 2D case, but this is not explicit.

The policy described in section 3.1 seems to be an incremental modification of existing approaches.  It is good that the paper clearly identifies what is different from previous work, but these differences seem relatively minor and the major conceptual innovation is not clearly described.

"We further crop the MR slices in k-space to the center (128 × 128) region."  This is not a reasonable thing to do, because it causes the complete loss of many of the clinically-interesting image features.  There is also no rationale provided for cropping, and no discussion of how practical issues are handled (e.g., like the fact that the fastMRI data is acquired with oversampled A/D, and data is given after analog filter and sampling but prior to digital filtering).

"Firstly, equispaced masks were shown by Hammernik et al. (2018) to be a strong handdesigned subsampling strategy for deep learning-reconstructed multi-coil MRI."  --> This again shows lack of familiarity with the MRI reconstruction literature.  Equispaced masks have been popular in MRI for 20+ years -- their usefulness and popularity was established in MRI decades before Hammernik's work.

The SSIM and NMSE numbers suggest that none of the reconstruction results shown in the paper have acceptable quality.  (It is impossible to evaluate the images themselves, which appear to have been compressed in the PDF).  Earlier optimal sampling approaches have achieved 4x acceleration with substantially lower errors than those reported in Table 2, which is a concern.

**Deanonymize Review:**

no

**Final Rating After The Rebuttal:**

2: Weak Reject

**Justification Of The Final Rating:**

The authors say that they have revised the manuscript, but the version I currently see in the system doesn't appear to reflect any of the changes they have described.  This makes it hard to change my final rating.

I also have remaining concerns about some of the authors' responses.  In the response, the authors defend their original descriptions as accurate.  I don't disagree that the original statements may have been factually correct, but it is possible to be factually correct and misleading/confusing at the same time.  Despite being factual, the authors' original descriptions are likely to be misinterpreted by most readers.

For example, the way the authors present Eq. 2 makes it seem like this is a well-accepted approach and that there are no other alternatives. The subsequent statement acknowledging that Eq. 2 leads to blurring would mislead many readers to believe that this blurring is a long-standing unsolved problem.  This will be misleading and confusing for readers, because this is not a well-accepted approach and there are many alternatives that do not result in blurring (e.g., the references by Sodickson, Pruessmann, Griswold, cited in the same paragraph).

As another example, the authors strongly emphasize that considering adaptive-sampling in the multi-channel setting with deep reconstruction is novel.  While this is factually correct, it is still misleading/confusing if there is no acknowledgment of existing methods that have considered optimal (but non-adaptive and without deep recon) multi-channel sampling and shown that it has different characteristics from optimal single-channel sampling.  (See the Haldar and Kim reference raised earlier, or the work by Gozcu et al in EUSIPCO 2019).  Or if there's no acknowledgment of Seeger's work that considers adaptive sampling for non-deep reconstruction, and which would be trivial to generalize to the multi-channel setting.

I also think it's important for the paper to clearly acknowledge that artificially reducing the spatial resolution of the images makes the results of the paper irrelevant to real MRI applications.  I understand why the authors chose to do this and would have no problem if the authors clearly demarcated it as a limitation of the study, but the failure to disclose a major issue like this is going to be misleading/confusing to many readers.

Finally, I would disagree with the sentiment that it is impossible to perform direct comparisons between different MRI papers.   Metrics like NMSE or NRMSE  have no user-selected parameters and measure basically the same thing as PSNR -- these would be much better metrics to report if the authors are worried about this issue.    But the real issue, which the authors have not addressed, is that the quantitative metrics the authors report are surprisingly low compared to what the MRI community views as acceptable numbers, and the image examples the authors show are missing clinically-relevant features.  These are again limitations that should be disclosed to avoid misleading/confusing readers.

**Paper Type:**

methodological development

**Questions To Address In The Rebuttal:**

The literature review needs to be substantially improved, and the problem context needs to be described more accurately.  The issues about image quality and realism also need to be addressed, as these are major limitations.

**Special Issue:**

no

---

### Official Review · Reviewer_QgqC · 2022-01-24

**Confidence:** 4
**Preliminary Rating:** 3
**Recommendation:** Poster

**Summary:**

The paper proposes an adaptive sub-sampling policy network for multi-coil compressed sensing MRI. The method builds up on E2E-varnet, and ideas from prior works on learning the sampling patterns to build an end-to-end system for both learning to sample, and reconstruct.
The authors show some interesting findings: non-adaptive policies may perform better; vanilla equidistant policies may perform better than LOUPE in the parallel imaging setting.

While I find the paper an interesting read, I am not completely convinced of the novelty of this paper. Mainly, the paper discusses development of learnable sampling strategy in the **multi-coil setting**, saying that most prior works focused on the **single-coil setting**. Currently, I do not see the differences other than the implementation details. Does the incorporation of sensitivity maps complicate the process? If so, what efforts were made to overcome this hardship?

**Strengths:**

1. The proposed strategy clearly outperforms LOUPE, and equidistant sampling strategies.

2. It is appealing that the authors built upon E2E-varnet, the current SOTA, to show the improvements. Other works mainly used simpler methods, such as vanilla U-Net.

3. Some interesting insights were discussed, especially that the learnt policy better performs with the non-adaptive strategy. For the reasoning, I do not find the current hypotheses so convincing. However, this could be developed in future works.

**Weaknesses:**

1. As stated in the summary, I am not convinced why converting SC --> MC setting leads to a novel research paper. If I am wrong, I would like to kindly ask the authors to emphasize on the important differences and the difficulties.

2. Since the paper is mostly methodological development, the key part is the section **3.Methods**. However, I found the methods section very confusing to follow. Rather than repeating phrases such as "A part of our model was adopted from B paper.", it would be much more easy to follow if the authors emphasized on the *reason* of choosing such designs, with relevant equations and figures (diagrams).

3. Inspecting the results figure, I observe that all of the MR images contain folding artifacts along the vertical axes. This is not desirable, and can hamper the strength of the paper by quite a margin. I would advise the authors to re-prepare the dataset, eliminating such artifacts. Center-cropping should not be done in k-space. It should be performed in the image space.

**Deanonymize Review:**

no

**Final Rating After The Rebuttal:**

4: Weak Accept

**Justification Of The Final Rating:**

The main novelties of the paper were much clarified, although I would expect some readers to be just as confused. However, the authors promised to open-source their code, and by that such issue will be alleviated. Moreover, it now seems that the data pre-processing issue is resolved. I raise my rating to 4.

**Paper Type:**

both

**Questions To Address In The Rebuttal:**

1. Main novelties of the paper should be strengthened.

2. The differences in developing sampling strategy in the single-coil setting, and the multi-coil setting.

3. Preparation (pre-processing) of the experimental data.

**Special Issue:**

no

---

### Official Review · Reviewer_Bruw · 2022-01-24

**Confidence:** 5
**Preliminary Rating:** 4
**Recommendation:** Poster

**Summary:**

This paper jointly optimizes an adaptive sampling strategy and a reconstruction model, the End-to-end Variational Network. This is done for 1D-sampling in fastMRI multi-coil knee data. The best performing policies learn to be non-adaptive and lead to higher SSIM-scores than conventional approaches, particularly at high acceleration factors.

**Strengths:**

This paper is written in a comprehensive and structured way. Methodologically, the authors challenge earlier observations that sampling patterns must be optimized on a per-scan basis to lead to most optimal result, which is an interesting point-of-view in ongoing discussions on this topic.

Relevant benchmark experiments using LOUPE as deterministic random sampling method are performed.

**Weaknesses:**

As a preprocessing step, data has been cropped to a center 128x128 region. In all resulting images, an aliasing pattern can be seen along the frequency-encoding vertical axis, suggesting incorrect data preprocessing. The target also appears more blurred than it should. This is expected to also have affected the coil sensitivity estimation. It it unclear how this would affect the presented results. At worst, the introduced blur/ghosting, being deterministic, steers the non-adaptivity of the learned sampling.

**Deanonymize Review:**

no

**Detailed Comments:**


Minor:
Section 4.2 starts by summarizing the original equidistant sampling. This may in reality have been more practically driven, i.e., equidistant sampling being the only measurement method known at the time.
Currently, it is known that random sampling, e.g. deterministically through Poisson sampling leads to a better distribution of the aliasing noise and hence a sharper initial PSF.

**Final Rating After The Rebuttal:**

5: Strong Accept

**Justification Of The Final Rating:**

Thank you for addressing my comments. In particular, I am satisfied with the preliminary revised results you shared after adjusting the data preprocessing, which significantly improves image quality. The authors are right about the 1D sampling strategy applied to this data.

**Paper Type:**

methodological development

**Questions To Address In The Rebuttal:**

With improved preprocessing and aliasing/ghosting artifacts removed, do the conclusions of the paper still stand? Or will the proposed method then adapted to the data at hand?

The clinical relevance may be modest if the added value is only seen at really high acceleration factors of 8x. Please comment.

**Special Issue:**

no

---

### Meta-Review · Area_Chair_i93a · 2022-02-19

**Recommendation:** Accept (Poster)
**Confidence:** 3

**Metareview:**

The reviewers appreciated the paper, with an interesting topic and some interesting ideas.

Overall there are some concerns with some still lingering, especially with the novelty, processing effects, and proper characterization of the paper. Out of this, the most important to me seems to be the last one -- properly representing the context of the paper, the limitations, and the effects of the decisions taken (e.g. the cropping). This was alluded to by all reviewers, and especially thoroughly looked at the by reviewer rHh8. I agree with these concerns. I think the authors' response makes sense, for example in that it is not tackling the full realism of the problem and more the concept -- such ideas are still valuable even if not ready for actual MR implementation, but these limitations or caveats need to be made explicitly clear.

I think the paper should be accepted to enable discussion of the presented ideas at the conference since there is sufficient interest, *but* also believe the authors must address the concerns, especially those of reviewer 3 that are still lingering after the overall author response. Most of the concerns are in being clear in the paper about the limitations, connection to existing literature and context, etc.

---

### Decision · Program_Chairs · 2022-02-28

Accept